# A phase transformable ultrastable titanium-carboxylate framework for photoconduction

Sujing Wang[1,2], Takashi Kitao[3,4,11,12], Nathalie Guillou[2], Mohammad Wahiduzzaman[5],
Charlotte Martineau-Corcos[2,6], Farid Nouar[1,2], Antoine Tissot[1,2], Laurent Binet[7], Naseem Ramsahye[5],
Sabine Devautour-Vinot[5], Susumu Kitagawa[3,8], Shu Seki[9], Yusuke Tsutsui[9], Valérie Briois[10],
Nathalie Steunou[2], Guillaume Maurin[5], Takashi Uemura[3,4,11,12] & Christian Serre[1,2]

Porous titanium oxide materials are attractive for energy-related applications. However, many suffer from poor stability and crystallinity. Here we present a robust nanoporous metal–organic framework (MOF), comprising a $Ti_{12}O_{15}$ oxocluster and a tetracarboxylate ligand, achieved through a scalable synthesis. This material undergoes an unusual irreversible thermally induced phase transformation that generates a highly crystalline porous product with an infinite inorganic moiety of a very high condensation degree. Preliminary photophysical experiments indicate that the product after phase transformation exhibits photoconductive behavior, highlighting the impact of inorganic unit dimensionality on the alteration of physical properties. Introduction of a conductive polymer into its pores leads to a significant increase of the charge separation lifetime under irradiation. Additionally, the inorganic unit of this Ti-MOF can be easily modified via doping with other metal elements. The combined advantages of this compound make it a promising functional scaffold for practical applications.

[1] Institut des Matériaux Poreux de Paris, FRE 2000 CNRS, Ecole Normale Supérieure, Ecole Supérieure de Physique et de Chimie Industrielles de Paris, PSL Research University, 75005 Paris, France. [2] Institut Lavoisier de Versailles, UMR 8180 CNRS, Université de Versailles Saint-Quentin-en-Yvelines, Université Paris-Saclay, 78035 Versailles, France. [3] Department of Synthetic Chemistry and Biological Chemistry, Graduate School of Engineering, Kyoto University, Katsura, Nishikyo-ku, Kyoto 615-8510, Japan. [4] CREST, Japan Science and Technology Agency (JST), 4-1-8 Honcho, Kawaguchi, Saitama 332-0012, Japan. [5] Institut Charles Gerhardt, Montpellier UMR 5253 CNRS ENSCM UM, Université Montpellier, CEDEX 05, Montpellier 34095, France. [6] CEMHTI, UPR 3079 CNRS, CEDEX 2, Orléans 45071, France. [7] IRCP, Chimie ParisTech, 75005 Paris, France. [8] Institute for Integrated Cell-Material Sciences (WPI-iCeMS), Kyoto University, Yoshida, Sakyo-ku, Kyoto 606-8501, Japan. [9] Department of Molecular Engineering, Graduate School of Engineering, Kyoto University, Katsura, Nishikyo-ku, Kyoto 615-8510, Japan. [10] Synchrotron SOLEIL-UR1, 91192 Gif-Sur-Yvette, France [11]Present address: Department of Applied Chemistry, Graduate School of Engineering, The University of Tokyo, 7-3-1 Hongo, Bunkyo-ku, Tokyo 113-8656, Japan. . [12]Present address: Department of Advanced Materials Science, Graduate School of Frontier Sciences, The University of Tokyo, 5-1-5 Kashiwanoha, Kashiwa, Chiba 277-8561, Japan. These authors contributed equally: Sujing Wang, Takashi Kitao. Correspondence and requests for materials should be addressed to T.U. (email: uemura@sbchem.kyoto-u.ac.jp) or to C.S. (email: christian.serre@ens.fr)

Titanium oxide (Ti–O) compounds represent a well-known family of materials that are widely involved in current human life. Titanium dioxide ($TiO_2$), the most recognized member of this family, which features low toxicity, high natural abundance, and remarkable stability not only plays an essential role as a common component in everyday items (toothpaste, paint, sun cream, food pigment, plastic, rubber, etc.), but also has significant potential to contribute to the advanced fields related to energy systems (photocatalysis and solar cells)[1–3] owing to its promising electrochemical properties[4, 5]. Introduction of porosity into the $TiO_2$ structure is an efficient way to tune or improve its current performance by increasing accessible active sites and the diffusion of reactants and products, while opening the new possibility of its application in other domains[6]. Unfortunately, the recently discovered porous $TiO_2$ is limited by a decrease in the crystallinity and stability of the resulting materials, which strongly reduces the utility of its favorable properties[7]. Similarly, other porous crystalline Ti–O solids, such as Ti-silicates and Ti-phosphates, suffer from insufficient Ti content[8] and a low accessible porosity[9]. Therefore, the discovery of novel porous Ti–O-based materials with highly ordered crystalline structures, integrating the properties of existing Ti–O compounds with those of ordered porous materials, opens new horizons in health and sustainable applications.

In this context, the field of metal–organic frameworks (MOFs) may be able to offer a tentative solution to the aforementioned limitations. MOFs are three-dimensional (3D) hybrid solids composed of metal ions or clusters interlinked by organic molecules[10]. The unprecedented chemical and structural diversity displayed by MOFs has led to the development of highly porous architectures, paving the way toward potential applications such as separation, gas storage, and catalysis[11–13]. However, the field of Ti-MOFs is still in the early stage of its development, mainly due to the challenge of controlling titanium chemistry in solution. Very few examples of crystalline Ti-MOFs obtained from direct synthesis have been reported[14] since the first example was discovered in 2006 (MIL-91)[15]. None of the porous Ti-MOFs could meet the practical application requirements owing to several drawbacks, such as costly precursors, complex syntheses, toxic non-scalable reaction conditions, poor chemical stability, and limited porosity[16–24]. In addition, as the inorganic building blocks of Ti-MOFs are either discrete Ti ions, Ti-oxoclusters, or corner-sharing chains of Ti octahedra, the corresponding structures tend to show a low ratio of oxo-groups to Ti(IV) ions (oxo/Ti ≤ 1). In titanium chemistry, the oxo/Ti ratio of a Ti–O compound (also known as the condensation degree) is a critical parameter for the evaluation of its properties compared to those of $TiO_2$[25]. The higher this ratio, the closer the behavior of the material is to that of $TiO_2$. The focus is thus on the discovery of new Ti-MOFs that feature a highly ordered, crystalline, porous structure whose constituents are Ti–O-based inorganic units with a high condensation degree (>1) and long-term stability. For practical applications, these materials should also be prepared under scalable conditions using simple and less harmful starting regents.

We report herein a porous 3D mdip-based Ti-MOF, namely MIL-177-LT (MIL stands for Materials from Institut Lavoisier, LT for low temperature form and mdip for 3,3′,5,5′-tetra-carboxydiphenylmethane), also denoted MIP-177-LT (MIP stands for Materials from Institute of Porous Materials of Paris). MIL-177-LT, with the formula $Ti_{12}O_{15}(mdip)_3(formate)_6$, was prepared under mild, scalable reaction conditions by simply refluxing a mixture of $Ti(iPrO)_4$, $H_4mdip$ and formic acid. Its honeycomb crystal structure, which features nano-sized pores, is composed of a $Ti_{12}O_{15}$ cluster secondary building unit (SBU) with a high condensation degree of 1.25, surpassing that of all

previously reported Ti-MOFs. Furthermore, MIL-177-LT can survive under extremely acidic conditions, including aqua regia and concentrated $H_3PO_4$. One can also introduce, through a direct synthesis, significant amounts of iron(III) into this structure with adjustable ratios and homogeneous distributions, in order to tune the electrical conductive behavior of the solid. It is worth noting that a thermally induced phase transformation of MIL-177-LT to its high-temperature (HT) form, MIL-177-HT, leads to the formation of a one-dimensional (1D) infinite Ti–O subunit $(Ti_6O_9)_n$ with an even higher condensation degree of 1.5, while maintaining its highly crystalline porous architecture. This high-temperature form displays a strongly enhanced photo-responsive ability, as compared to MIL-177-LT, and far exceeds the performance of other discrete SBU-based Ti-MOFs, such as MIL-125[16]. In addition, the introduction of conductive polythiophene into its pores gives rise to an enhanced charge separation lifetime upon irradiation. To the best of our knowledge, this serves as the first example in porous materials, MOFs or otherwise, that demonstrates the alteration of the dimensionality of the inorganic building unit as an efficient way of tuning the physical properties.

## Results

**Overview**. Essential features of this work are described in Fig. 1, including the thermally induced structural transformation from the MIL-177-LT structure to the MIL-177-HT one with the corresponding increase of the condensation degree (oxo/Ti ratio) from 1.25 to 1.5, the fabrication of MOF/polymer composite to enhance the photoconductive properties of MIL-177-HT and the doping with other metal species into the inorganic unit of MIL-177 to modify its physical and chemical properties.

**Synthesis and characterization**. MIL-177-LT was synthesized by refluxing a mixture of $Ti(iPrO)_4$ and $H_4mdip$ in formic acid under ambient pressure (Supplementary Methods), generating hexagonal rod product particles with a uniform size distribution (Fig. 2a). Activation was achieved through a simple wash with ethanol at room temperature. The preparation and activation steps involve simple processes and less harmful chemicals than those used for the syntheses of other reported Ti-MOFs, allowing for facile scale-up. For instance, 100 g of MOF product can be obtained from a single reaction in a 2L round bottom flask. The structure of MIL-177-LT, determined by combining high-resolution powder X-ray diffraction (PXRD) data (Fig. 2b, Supplementary Table 1, and Supplementary Note 1) and density functional theory (DFT) calculations (Fig. 2c, Supplementary Table 2, and Supplementary Note 2), crystallizes in a hexagonal space group $P6/mmm$ with unit cell parameters of $a = b = 22.5943(4)$ Å, $c = 12.3060(3)$ Å, and $V = 5440.6(2)$ Å$^3$. Its $Ti_{12}O_{15}$ cluster SBU consists of a central cyclic hexamer of Ti octahedra capped above and below by corner-sharing trimers of Ti octahedra (Fig. 2d). It is strongly related to a previously reported molecular Ti-oxocluster[26] with a condensation degree of 1.25, which is higher than that of all other reported Ti-MOFs. Adjacent SBUs are linked by both formate ions and mdip linkers. Each mdip ligand connects four Ti-oxoclusters. Formates play two roles: the first is to bridge pairs of adjacent Ti ions in the cyclic hexamer, pointing toward the pore interspace; the second is to ensure the linkage of SBUs along the $c$-axis (Fig. 2e). The resulting 3D pore system features large (ca. 1.1 nm) accessible hexagonal channels running along the $c$-axis, as well as very narrow (ca. 0.3 nm) channels along the $a–b$ plane, characteristic of a typical $bnn$ topological network (Fig. 2f–i). Nitrogen porosimetry gives a Brunauer-Emmett-Teller (BET) area of 730(10) m$^2$ g$^{-1}$ and a free pore

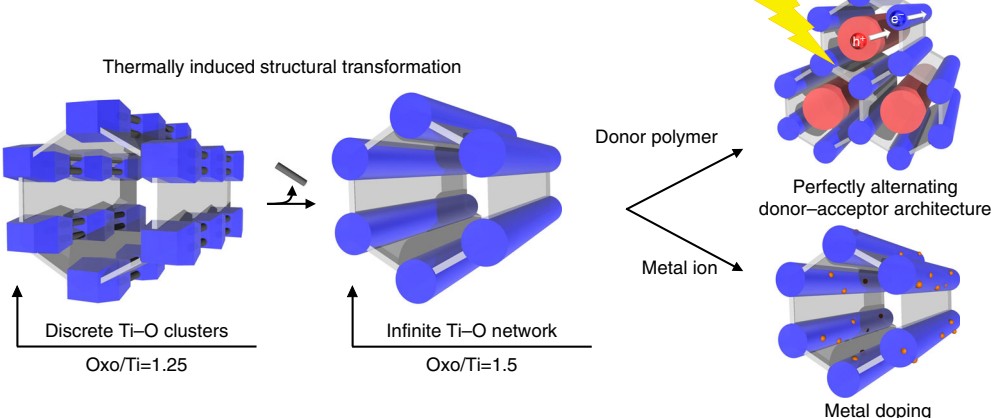

Thermally induced structural transformation

Donor polymer

Perfectly alternating
donor–acceptor architecture

Discrete Ti–O clusters
Oxo/Ti=1.25

Infinite Ti–O network
Oxo/Ti=1.5

Metal ion

Metal doping

**Fig. 1** Structural transformation of and modification to MIL-177 MOFs. A thermally induced structural transformation of the discrete $Ti_{12}O_{15}$ cluster SBUs linked by bridging formates in MIL-177-LT structure (left) to 1D infinite $(Ti_6O_9)_n$ nanowires in MIL-177-HT structure (middle) is observed with an increase of the condensation degree from 1.25 to 1.5. Introduction of conductive polythiophene into the pores of MIL-177-HT gives rise to a perfectly alternating donor–acceptor architecture for photoconduction application (right top). Other metal species can be doped into the inorganic unit of MIL-177 to introduce modification to its properties (right bottom)

volume of 0.47(8) $cm^3 g^{-1}$ (Supplementary Fig. 6), consistent with the theoretical values of 700 $m^2 g^{-1}$ and 0.47 $cm^3 g^{-1}$, respectively. The other characterization results, such as thermogravimetric analysis (TGA), temperature-dependent PXRD, elemental analysis, were included in the Supplementary Information (Supplementary Fig. 1–9, Supplementary Table 3, and Supplementary Note 3).

**Chemical stability**. MIL-177-LT remains intact in water at room temperature for months (Supplementary Fig. 10). However, exposure to boiling water leads to a progressive decrease of crystallinity during the first 12 h but does not further damage the structure afterward, possibly due to the formation of a very thin amorphous titanium oxide layer over the MOF surface, which could efficiently slow down the hydrolysis of MIL-177-LT particles. A similar trend was noticed under moderately basic aqueous conditions. Like other water-stable MOFs based on group IV metals, the framework of MIL-177-LT is easily destroyed by strong bases, such as $NH_4OH$ vapor. Nevertheless, MIL-177-LT possesses an excellent stability against extremely acidic conditions at room temperature, easily tolerating concentrated HCl (37%), $HNO_3$ (65%), and $H_2SO_4$ (98%); even aqua regia and concentrated $H_3PO_4$ (6 M) did not induce significant deterioration of the structure. Notably, no MOF compound that survives in such a high concentration of $H_3PO_4$ has been reported so far, since $H_3PO_4$ is extraordinarily destructive to MOF materials. As a result, only stability tests on MOFs at low concentrations of $H_3PO_4$ could be found[27, 28]. $NH_2$-MIL-125 shows a good water stability and was thus selected for comparison. In contrast to MIL-177-LT, $NH_2$-MIL-125 undergoes a much more rapid degradation under all the conditions tested.

Analysis by PXRD of MIL-177-LT samples, before and after treatment with acid, indicates that the crystal structure is largely unaffected despite slight changes in the relative intensities of some peaks (Fig. 3a). Spectra obtained from solid-state NMR demonstrated no substantial difference before and after the acid treatments, verifying that the coordination and connection of mdip linkers to the SBU remain unchanged (Supplementary Fig. 11). The porosity is also retained, although a slight reduction of nitrogen uptake is observed in some cases (Fig. 3b). This is likely due to the presence of residual acidic species trapped inside

the pores, as evidenced by SEM-EDX and FT-IR (Supplementary Table 4 and Supplementary Fig. 12). Interestingly, the sample exhibits even higher porosity after treatment with aqua regia compared to the as-synthesized compound. As previously observed for other metal(IV)-based MOFs, this could be due to a thorough cleaning of the initially inaccessible porosity upon this acidic treatment[29].

To the best of our knowledge, MIL-177-LT represents the first example of a carboxylate-based MOF that is resistant to the aforementioned extremely acidic conditions, especially aqua regia and concentrated $H_3PO_4$. This is clearly a leap forward in terms of chemical stability, not only for Ti-MOFs, but also for the entire carboxylate-based MOF subclass.

**Modification of the inorganic moiety in the MIL-177-LT structure**. It is worth noting that the MIL-177-LT structure possesses a large potential for further modification to the inorganic $Ti_{12}O_{15}$ cluster moiety. The first possibility involves the terminal formates that are pointing toward the nano-sized channel. These groups can either generate unsaturated metal sites upon the departure of the terminal species, or undergo exchange reactions with functional groups or molecules, such as hydroxyls, water, carboxylic acids, sulfonic acids, and phosphorous acid derivatives, as reported previously for the other metal-oxocluster-based MOFs[30–32]. A second possibility arises from the introduction of metal elements, including transition metals (V, Cr, Mn, Fe, Co, Ni, Cu, Ru, Nb…) and main group metals (Sn, In…), that can easily replace Ti atoms in the SBU. This can be achieved by direct synthesis with the intended metal, with adjustable ratios and homogeneous distributions. To that end, an iron-doped MIL-177-LT sample was prepared (Supplementary Methods) and the homogeneous introduction of ca. 11% Fe(III) (atomic ratio based on Ti) was confirmed through a combination of advanced characterizations, namely energy-dispersed X-ray spectroscopy, magnetic measurements, electron paramagnetic resonance, and extended X-ray absorption fine structure (Supplementary Figs. 26–34, Supplementary Tables 5–8, and Supplementary Notes 5 and 6). This is of particular importance in the tuning of the chemical and physical properties of MIL-177-LT as it is well documented that doping with other elements is a powerful strategy to enhance or modify the functionalities of bulk $TiO_2$[33, 34].

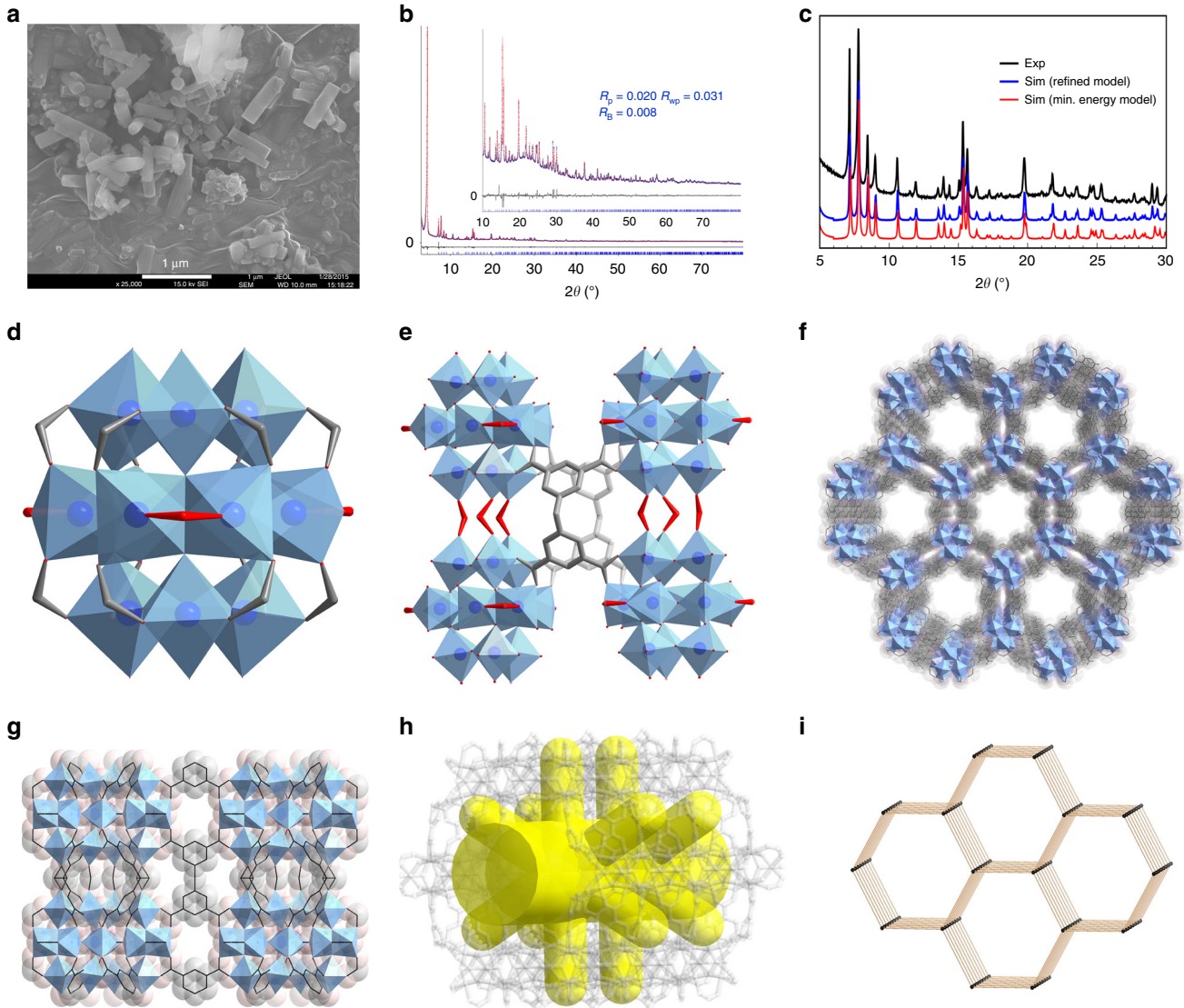

**Fig. 2** Structural characterization and detail of MIL-177-LT. **a** SEM image of MIL-177-LT particles with a hexagonal rod morphology and a uniform size distribution. **b** Final Rietveld plot of MIL-177-LT structure. **c** Comparison of the PXRD patterns obtained from experimental data, theoretically derived minimum energy structure and refined structure model of MIL-177-LT. **d** A $Ti_{12}O_{15}$ cluster SBU with 12 carboxylate groups from mdip linkers (in gray) and terminal formate groups (in red). **e** Adjacent $Ti_{12}O_{15}$ cluster SBUs with terminal and bridging formates (in red) connected by mdip linkers (in gray). **f** Nano-sized channels with a free diameter of 1.1 nm when viewed along the c-axis. **g** Small channels of 0.3 nm windows when viewed along the b-axis. **h** Overall pore shape of MIL-177-LT framework. **i** bnn topological network of MIL-177-LT

**A thermally induced irreversible phase transformation**. When MIL-177-LT is calcined at 280 °C for 12 h, it transforms irreversibly into MIL-177-HT (Fig. 4a). The structure has been determined and refined through PXRD (Fig. 4b) and DFT calculations (Fig. 4c). Remarkably, the terminal and bridging formates in the LT structure were removed, resulting in an unusual rearrangement of Ti–O bond connections: the discrete $Ti_{12}O_{15}$ clusters in MIL-177-LT were transformed into a 1D infinite $(Ti_6O_9)_n$ nanowire (Fig. 4d) with a condensation degree of 1.5, unprecedented among all Ti-MOFs. The nanowire is composed of edge-sharing trigonal pyramidal $Ti_6O_9$ clusters. The same hexamer is found in the $TiO_2$ anatase structure and a recently discovered bimetallic Ti–Cu-MOF[35]. The mdip linker adopts a similar coordination geometry as in the MIL-177-LT structure, connecting neighboring nanowires together to complete the 3D framework (Fig. 4e, f). These ultrathin Ti–O nanowires with a diameter of only ca. 1.0 nm are inaccessible using conventional inorganic syntheses, including sol-gel synthesis, thermal and

chemical bath deposition[36, 37]. Another detail worth mentioning is that the large hexagonal channels are retained in the MIL-177-HT structure with a slightly contracted diameter of ca. 0.9 nm, associated with a BET area of 690(10) $m^2 g^{-1}$ and a free pore volume of 0.45(5) $cm^3 g^{-1}$ (Supplementary Fig. 6). This is consistent with the theoretical nitrogen-accessible surface area of 800 $m^2 g^{-1}$ and free pore volume of 0.40 $cm^3 g^{-1}$, respectively.

To the best of our knowledge, an increase in the dimensionality of the inorganic building unit during structural phase transformation, such as connecting 0D clusters into 1D infinite chains or 1D chains into 2D layers through coordination bonds, is a universal observation for solid-state materials. Typically, however, the poor crystallinity of the corresponding product and the lack of efficient analytical methods result in difficulty with characterization. Consequently, very few examples have been reported, particularly for MOFs compounds[38–40]. In light of this, MIL-177-LT and HT represent an appropriate system of fundamental interest. MIL-177-HT can be obtained when doped with different

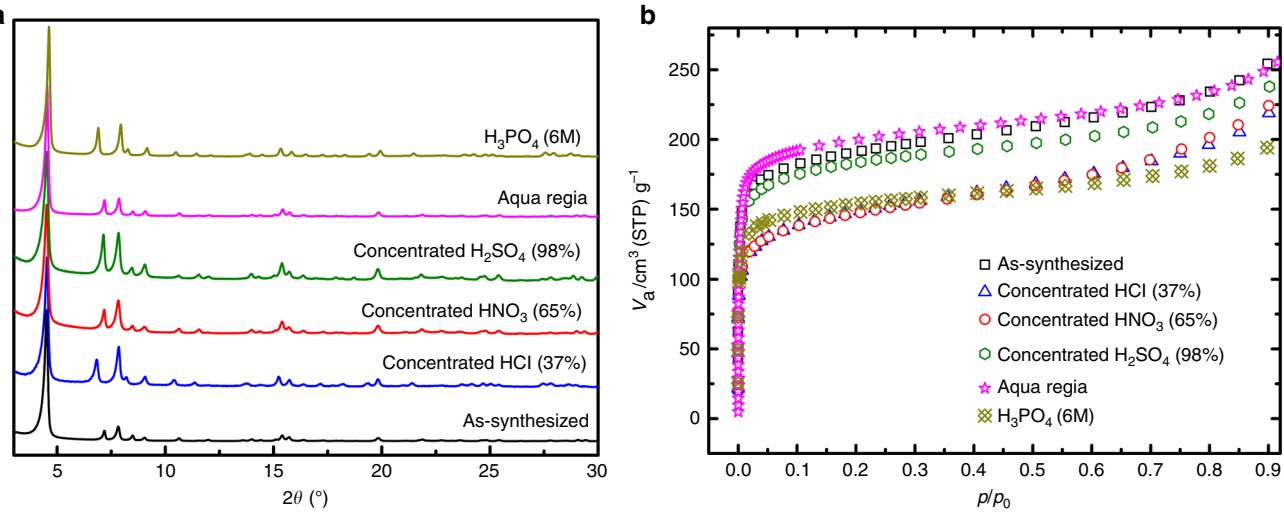

**Fig. 3** Chemical stability test results of MIL-177-LT. **a** PXRD patterns and **b** nitrogen adsorption isotherms of MIL-177-LT samples before and after chemical treatment in various acids

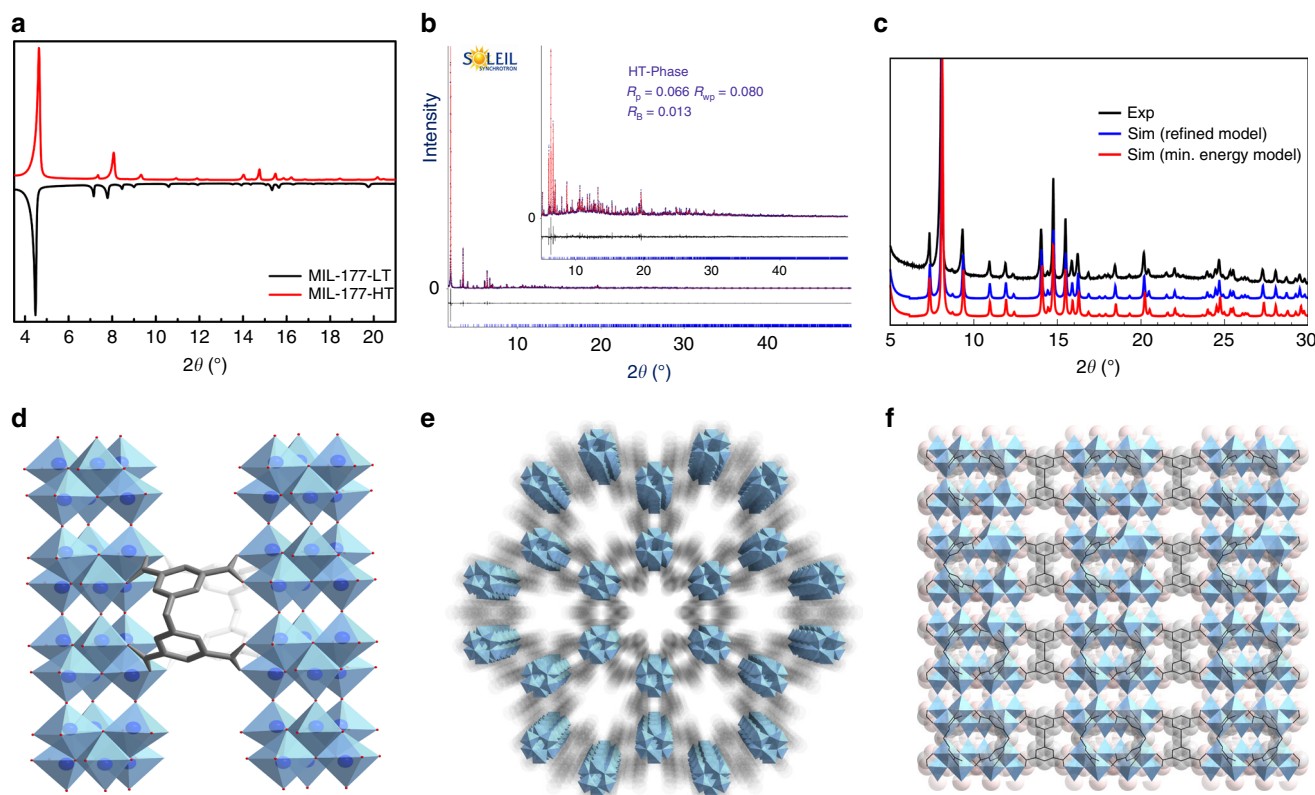

**Fig. 4** Structural characterization and detail of MIL-177-HT. **a** PXRD comparison between MIL-177-LT and HT compounds. **b**. Final Rietveld plot of MIL-177-HT structure. **c** Comparison of the PXRD patterns obtained from experimental data, theoretically derived minimum energy structure and refined structure model of MIL-177-HT. **d** Adjacent infinite ultrathin $(Ti_6O_9)_n$ nanowires with a thickness of ca. 1 nm connected by mdip linkers (in gray). **e** Channels between the $(Ti_6O_9)_n$ nanowires array running along the c-axis with a diameter of ca. 0.9 nm. **f** Small channels of 0.3 nm windows retained when viewed along the b-axis of the MIL-177-HT crystal structure

metals, leading to a family of HT compounds that comprise the doped Ti–O nanowire array in their structures. As a typical illustration, the electrical conductive performance recorded at 1 Hz and 373 K shows an increase of $5.2 \times 10^{-14}$ S cm$^{-1}$ to $1.5 \times 10^{-10}$ S cm$^{-1}$ upon Fe doping, nearly four orders of magnitude (Supplementary Figs. 26–34, Supplementary Tables 5–8, and Supplementary Notes 5 and 6). This change is most probably due

to the fact that the iron doping is expected not only to inject free charge carriers, but also to facilitate charge carrier movement.

**Photoconductivity**. To analyze the impact of the dimensionality change of the SBU in MIL-177 (LT: 0D; HT: 1D), the physical properties were studied from both experimental and theoretical

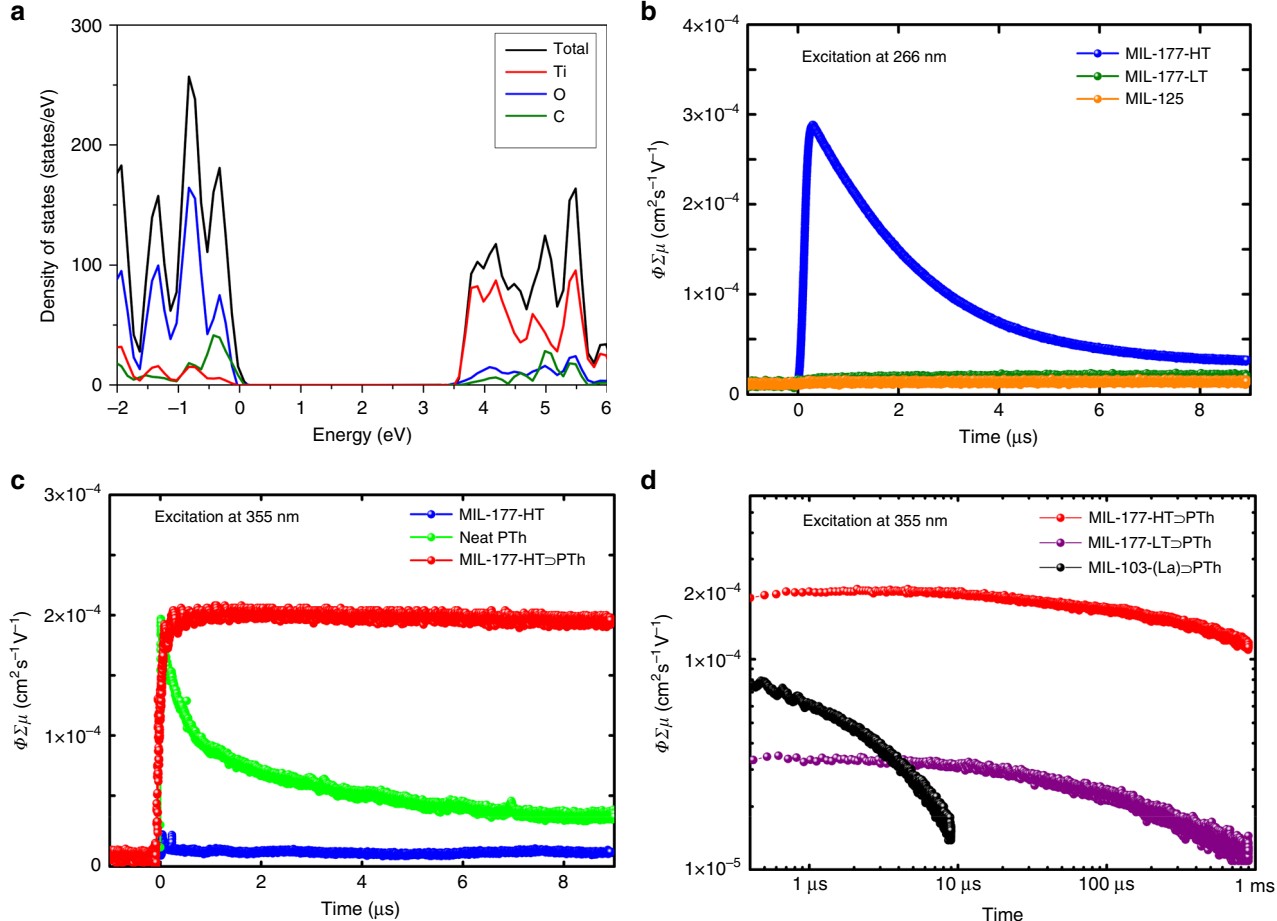

**Fig. 5** Physical properties of MIL-177 and related composites. **a** Total and partial PDOS of MIL-177-HT calculated at the PBEsol level to determine the contribution of the different orbitals. **b** Conductive transients observed by FP-TRMC upon excitation at 266 nm with a UV laser (MIL-177-HT in blue, MIL-177-LT in green and MIL-125 in orange); **c** Conductive transients observed by FP-TRMC under 355 nm near-UV irradiation (MIL-177-HT⊃PTh in red, MIL-177-HT in blue, and neat PTh in light green); **d** Conductive transients observed by FP-TRMC under 355 nm near-UV irradiation showing the extremely long lifetime of MIL-177-HT⊃PTh (>1 ms, in red) with MIL-177-LT⊃PTh in purple and MIL-103(La)⊃PTh in black for comparison

points of view. First, the diffuse reflectance spectra collected at 300 K revealed an experimental band gap of 3.67 eV for MIL-177-HT (Supplementary Fig. 14). This value is comparable to that of bulk anatase (3.2 eV)[41], nano-sized anatase (3.4–3.9 eV)[42], and other reported Ti-oxocluster-based MOFs such as MIL-125 (3.7 eV)[43] and COK-69 (3.8 eV)[21]. DFT calculations using the HSE06 hybrid functional confirmed these values: 3.5 eV for MIL-177-HT and 3.3 eV for anatase. In addition, further information on the electronic structures of both anatase and MIL-177-HT were obtained from the analysis of the projected density of states (PDOS) calculated using the PBEsol functional. For MIL-177-HT, the valence band mainly consists of states from the oxygen atoms, as well as a small contribution from the carbon atoms; the titanium atoms contribute more heavily to the conduction band, rather than the valence, as in the case of anatase (Fig. 5a and Supplementary Figs. 15 and 16).

Next, photophysical properties were analyzed through flash photolysis time-resolved microwave conductivity (FP-TRMC) measurements and compared with TiO$_2$ and another Ti-oxocluster-based MOF, MIL-125[44] (Fig. 5b). As expected, MIL-177-LT and MIL-125, both composed of discrete Ti–O cluster SBUs, generate extremely weak photoconductivity signals upon ultraviolet (UV) laser irradiation ($\lambda = 266$ nm) due to the lack of conduction pathways in their frameworks. In sharp contrast, a pronounced photoconductivity signal is observed for

MIL-177-HT under the same condition. The carrier mobility in MIL-177-HT was calculated to be at least $4 \times 10^{-4}$ cm$^2$ s$^{-1}$ V$^{-1}$ (Supplementary Fig. 17), which is comparable to those of well-known nano-sized TiO$_2$ materials[45, 46]. Recently, conductive MOFs have attracted considerable attention; however, this research field had mainly focused on the electrically conductive materials[47–51]. Until now, very few photoconductive MOFs have been documented and their photoconductivity mainly comes from photoactive organic linkers[52, 53]. Therefore, to the best of our knowledge, MIL-177-HT is the first MOF whose photoconductivity arises mainly from the inorganic infinite Ti–O building unit. This also strongly suggests that the photoconductivity of a coordination polymer can be tuned by increasing the dimensionality of its inorganic building unit.

In order to decrease the band gap of MIL-177-HT, thereby improving its photoactivity, one can follow the conventional strategy of combining TiO$_2$ and conjugated polymers[54]. As reported previously for other large pore MOFs, the nano-sized channel in MIL-177-HT makes it an ideal host system to accommodate polymer chains[55–57]. This could lead to a perfectly alternating donor/acceptor architecture at the molecular level between the accommodated polymer and MOF host, forming a large interface for charge separation and providing ambipolar pathways. To that end, the pores of MIL-177-LT (MIL-177-LT⊃PTh) and MIL-177-HT (MIL-177-HT⊃PTh) were filled with

polythiophene (PTh) to form composites (Supplementary Figs. 19–25) and the photoconductive performances of the materials were investigated (see "Methods" for details). In contrast to the original host, MIL-177-HT⊃PTh displays a clear photoconductivity signal when excited with a 355 nm laser beam due to the visible light sensitization with PTh (Fig. 5c). However, the photoconductivity of MIL-177-LT⊃PTh was found to be much lower than that of MIL-177-HT⊃PTh, highlighting the key role that the infinite Ti–O sub-network in MIL-177-HT plays in the facilitation of effective charge migration. An even more striking result was obtained from the analysis of the charge carrier lifetime. The charge carriers generated in MIL-177-HT⊃PTh under photoexcitation were extremely long lived. Individually, the host (MIL-177-HT) and neat PTh exhibited photoconductivity signals with a rapid decay; the half lifetimes ($\tau_{1/2}$) of the charge carriers in MIL-177-HT and neat PTh were calculated to be 2.2 μs and 0.7 μs, respectively (Fig. 5c). However, the lifetime for MIL-177-HT⊃PTh was remarkably extended into the millisecond range ($\tau_{1/2} = 1.0$ ms). The formation of the infinite Ti–O network in MIL-177-HT enabled the prolonged charge carrier lifetime, as revealed through the comparison with MIL-177-LT ($\tau_{1/2} = 100$ μs) and the electro-inactive MOF MIL-103(La)[58] ($\tau_{1/2} = 2.1$ μs) as host matrices (Fig. 5d). The observed carrier lifetime in MIL-177-HT⊃PTh is among the longest observed for solid-state dye-sensitized Ti–O systems[59]. These results suggest that the aligned bicontinuous conduction pathway accounts for the long-distance charge delocalization and the exceptional long-term charge retention. Our findings reveal that the hybridization of donor polymer and acceptor MOF provides the ideal architecture for charge separation, which appears to be a very promising model system for the understanding and further development of efficient photoenergy conversion devices. Nevertheless, due to the complexity of the MOF and composite system, more information about its photoconductive behavior, optimization of its performance, and detailed investigation are still needed to fully understand the mechanisms in play.

MIL-177-LT, a novel nanoporous titanium–carboxylate framework material, has been successfully prepared via an easily scalable synthesis using less harmful chemicals. The material features robust nano-sized porosity, a high condensation degree, and excellent chemical stability in extremely acidic conditions, as well as a platform for further functionalization through the inorganic building unit. Upon heating, an unusual irreversible phase transformation occurs forming MIL-177-HT, a porous crystalline Ti-MOF based on ultrathin Ti–O nanowires separated by mdip linkers. MIL-177-HT possesses the highest condensation degree in Ti-MOFs reported so far. Preliminary photoconductivity tests demonstrate the utility of the 1D infinite Ti–O nanowire SBU in MIL-177-HT, in comparison with the discrete SBU-based Ti-MOFs, producing a photoresponsive behavior comparable to that of bulk TiO₂. Finally, a MOF-conductive polymer composite MIL-177-HT⊃PTh displays considerable photoconductivity with a notably long lifetime as a result of its efficient alternating donor/acceptor architecture. MIL-177 not only outperforms all the other reported Ti-MOFs (e.g., NH₂-MIL-125, MOF-901, PCN-22, and COK-69) in terms of its simpler synthesis and excellent stability with considerable porosity, but also displays photoresponsive behavior close to that of TiO₂. Considering all the aforementioned characteristics, MIL-177 reveals itself to be a promising functional scaffold, paving the way for Ti-MOFs toward practical applications.

## Methods

**Typical synthesis of MIL-177-LT**. To a 25 mL round bottom flask, H₄mdip (200 mg, 0.58 mmol) and formic acid (10 mL) were added and stirred at room temperature until the solid dispersed uniformly. Then Ti(iPrO)₄ (400 μL, 1.32 mmol) was added dropwise, to avoid formation of large pieces of white precipitate. Afterward, the reaction mixture was heated under reflux for 24 h. After cooling to room temperature, the white solid product was filtered under reduced pressure and washed with ethanol. Large-scale synthesis (such as 10 or 100 g scales) can easily be achieved with this method.

**Typical preparation of MIL-177-HT**. The solid MIL-177-LT compound (200 mg) was ground into a fine powder, transferred to a Petri dish and dispersed uniformly, forming a thin layer. The MIL-177-LT powder was then heated to 280 °C for 12 h, forming the MIL-177-HT structure as a dark yellow powder.

**Preparation of MIL-177⊃PTh composites**. After drying the host MIL-177 compound (150 mg) by evacuation (<0.3 kPa) at 120 °C for 5 h in a Pyrex reaction tube, MIL-177 was immersed in a chloroform solution (1 mL) of terthiophene (17 mg) at room temperature. To incorporate terthiophene into the nanochannels, chloroform was completely removed by evacuation at 100 °C for 3 h to obtain MIL-177-terthiophene composites (MIL-177⊃TTh). MIL-177⊃TTh was mixed with iodine (43 mg), and then heated at 120 °C for 48 h to facilitate the oxidation polymerization and afford MIL-177-polythiophene composites (MIL-177⊃PTh). Upon cooling, the solids were dispersed in methanol (50 mL) and collected by centrifugation and subsequently washed with 5% v/v hydrazine monohydrate in ethanol (20 mL) for dedoping. The solids were then collected by centrifugation and washed with ethanol (20 mL) and water (20 mL) prior to drying under vacuum at 120 °C overnight.

**Data availability**. Further experimental and computational details can be found in the Supplementary Information. X-ray crystallographic data for the structures of MIL-177-LT and MIL-177-HT can be found in Supplementary Data 1 and 2. All relevant data supporting the findings in this work are available from the corresponding authors on request.

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

## Acknowledgements

S.W., N.G., C.M.-C., F.N., A.T., N.S., and C.S. acknowledge the financial support of the European Community within the Seventh Framework Programme (FP7) under Grant Agreement No. 295775 (Project SoTherCo). T.K. and T.U. acknowledge the financial support from CREST-JST and JSPS KAKENHI Grant Number JP16H06517 (Coordination Asymmetry). M.W., N.R., S.D.-V., and G.M. thank Institut Universitaire de France for its support and the financial support of ANR-13-SEED-0001-01 "Chesdens" project. Measurements at the ROCK synchrotron beamline of SOLEIL was supported by a public grant overseen by the French National Research Agency (ANR) as a part of the "Investissements d'Avenir" Program (ANR-10- EQPX-45). The EPR measurement was supported by IR Renard, FRE CNRS 3443. The authors thank Erik Elkaim for the access to the CRISTAL synchrotron beamline of Soleil. The authors acknowledge Victoria Steggles and Hazelle Wang for their valuable effort in editing the manuscript.

## Author contributions

S.W. contributed to the synthesis and characterization of the Ti-MOFs and the Fe-doped samples, and participated in the writing of this article. N.G. solved the crystal structures from X-ray diffraction powder data. C.M.-C. carried out the solid-state NMR characterizations. F.N. contributed to the preparation and characterization of the Ti-MOF samples. A.T. performed the magnetic and EPR measurements. N.S. helped the synthesis of the Ti-MOFs and the writing of this article. C.S. was the coordinator of the whole study and led the writing of the article as well as closely supervising the synthesis and characterization of the work. L.B. coordinated the EPR measurement. V.B. carried out the EXAFS characterization and analysis. M.W. was in charge of the DFT structure optimization calculations. S.D.-V. performed the conductivity experiments. N.R. did the computational analysis of the electronic properties of the MOFs while G.M. supervised all the computational studies as well as participated to the writing of the article. T.K. carried out the experimental work related to the polymer loading into the MOFs as well as performing the photophysical characterizations with Y.T. S.S. supervised the photophysical experiments. S.K. contributed to the writing of the article. T.U. supervised the

polymer-MOF composite-related experimental work as well as contributing to the writing of the article.

## Additional information

**Competing interests:** The authors declare no competing interests.

