## [Peer Review File · Nature Communications]

Reviewers' comments:

Reviewer #1 (Remarks to the Author):

Review report for NCOMMS-17-32492, by C. Serre and co-workers.

The present paper is very well written and presents two new Ti-based MOF structures obtained from Ti alkoxides. These MOFs are very resistant to aggressive chemical media, they are compact and able to include host species or doping metal-species. The most interesting point of this paper in my opinion, is the less explored one: photoconductivity of the poly(thiophene) intercalates, since they achieved long-lived charge carriers upon excitation with light.

The paper is concise, coherent and data were correctly analyzed and clearly presented.

Experimental procedures are coherent with and adequate for this kind of study. This referee could not identify methodological flaws that would prohibit its publication in Nat. Commun. The authors came to several conclusions, and they are clearly supported by the results they have presented. In spite of all these positive points, I can't consider this paper as highly innovative and original.

The main points presented in the paper are not new (except for the structure mainly), and don't bring light over unsolved questions. Alkoxide/carboxylate Ti MOFs are not new, and I could not understand why the authors mention the field (l. 73) is still in its infancy...there are plenty of papers about the subject, since 10 years at least, as mentioned in the paper. The statement has to be reconsidered.

The authors also insist on using the "green chemistry" argument. Sincerely, this is far from being green! For example, (l. 113, 115) formic acid is not green at all. It was used as a solvent and a building block (in excess- supp info l. 184). It is an inhibitor of the mitochondrial cytochrome oxidase causing histotoxic hypoxia. (Pharmacol Toxicol. 1991 Sep;69(3):157-63.)

The authors criticize (l. 77) other Ti MOFs for being "non-green" as I could understand. Most use other alkoxides, and those used in the present paper are not better. The authors should not emphasize green aspects, but the remarkable stability.

Ti alkoxides are quite difficult to produce in scale, and honestly I don't think that Ti(iPrO)₄ is cost effective (l. 92); maybe when compared to other Ti compounds, but not in a general way, for industrial purposes. Is it cheaper than TiCl₄?

The references are outdated. Many important or new papers in the field should be cited as for example : New J. Chem., 2017, 41, 14030 / C. K. Brozek and M. Dinca, J. Am. Chem. Soc., 2013, 135, 12886-12891. / J Am Chem Soc. 2016 Apr 6;138(13):4330-3, to cite a few. This field is growing fast.

There are some minor issues too:

a: (l. 69) materials (misspelled)

b: The authors should comment the main differences between their MOFs and MOF 901 from Yaghi et al.

c: (l. 144) what about resistance to bases? (ammonia especially). Some figure in FIG S12, why nothing is mentioned in the main text about bases?

d: Is there any reference for MIL-103 (la)?

For the reasons above, I would not recommend it for publication in the present form.

Reviewer #2 (Remarks to the Author):

This article presents synthesis of a new MOF consisting Ti-O framework and its thermally induced phase transition to form highly crystalline porous material. This material has a similarity with TiO₂ and exhibits high chemical stability, especially against strong acid, and photoconductivity. The detailed study on these new titanium MOFs is sophisticated and impressive. I judge this article worthy for publication in Nature Communications, if the following points are reconsidered.

1) Lines 210-211: The reason why the conductivity increase upon Fe doping should be explained.

2) Lines 214-216: It is described that the band gap of 3.67 eV for MIL-177-HT is close to that of

anatase (3.2 eV), but the difference between them 0.5 eV is not small.

Reviewer #3 (Remarks to the Author):

The manuscript by Uemura, Serre and coworkers reports two new Ti-O based solids that can be generated using relatively facile synthetic methods. Ti-O based materials are widely used throughout society, and in this regard, the development of new systems is highly important. I address first specific concerns with the manuscript.

- 1) A number of sentences throughout the manuscript are unclear, and require clarification. For example, in the introduction "Unsatisfactorily, current progress on..."; "...can offer a new line of thoughts."; "These materials.."; "have been hindered away from..."; "...no matter the inorganic building blocks of those Ti-MOFs are discrete Ti ions..." In general, the grammar requires attention throughout this manuscript.
- 2) References 11, 12 and 13 in the introduction do not appear to be representative of the "potential applications" mentioned. The authors should consider instead referencing one or two very recent, up-to-date, and definitive books or comprehensive reviews in the field. Readers outside the field need to be able to refer to literature that will provide an overarching view of the area.
- 3) A reference is missing for "MIL-125" on page 2 of the introduction.
- 4) In general, the authors should specifically refer to the figure/table in the supplementary information that is being referred to. A couple of supplementary figures are noted throughout the article, but these do not appear to be in any particular order.
- 5) In the Synthesis and Characterization, the authors should direct the reader to the data demonstrating "uniform size distribution" (this should also be mentioned in the main text), and "scale-up to any quantity levels" (in addition to rewording this text, mention should be made of just how scalable the synthesis is).
- 6) In Chemical Stability, the authors should again direct the reader to data that confirms the stability under the various conditions discussed.
- 7) Defects are critical in framework chemistry with regard to functional properties such as gas storage and optical/electronic properties (eg, photoconductivity). Under Chemical Stability, the authors mention that there may be structural defects. Were these investigated and characterized? The measured properties are for the samples with the defects generated under the synthetic conditions trialled, so the inclusion of such information is important.
- 8) Under Figure 4, "To the best of our knowledge, increase of dimensionality of the..." is unclear and needs revision.
- 9) In the Photoconductivity section, the measured conductivities of <math><10^{-10}</math> S/cm are in the insulating domain. Thus, the authors should be very cautious in their use of "electrical conductivity" and "photoconductivity". In the abstract, the authors mention the improvement in "electrical conductive performance" with doping, however it is the photoconductive performance that is tuned.
- 10) In the Photoconductivity section, "Secondly, first..." should be fixed.
- 11) References 42, 43 and 44 in the Photoconductivity section are for electrical conduction in MOFs, not photoconductivity. The authors need to instead reference specific works in the MOF field that report photoconductivity.
- 12) Supporting Information Figure S2 – what is the origin of the indentation at ~450 deg C in the plot for MIL-177-HT?

The discovery here is of widespread importance, however the manuscript requires major revision. I recommend that the manuscript be considered for publication subject to all reviewers concerns being addressed.

Response to Reviewers' comments:

Reviewers' comments:

Reviewer #1 (Remarks to the Author):

Alkoxide/carboxylate Ti MOFs are not new, and I could not understand why the authors mention the field (l. 73) is still in its infancy...there are plenty of papers about the subject, since 10 years at least, as mentioned in the paper. The statement has to be reconsidered.

Response

As the referee pointed out, the research on the field of Ti-MOF started more than ten years ago and there are considerable number of papers dealing with this topic. In comparison with the other metals based MOFs, the overall number of Ti-MOFs is still very limited (less than 30 including compounds obtained in direct synthesis and post exchange methods). Following the referee's comment, we have already changed the statement as 'it is still in the early stage of its development' in the revised manuscript (line 72) and marked it in red.

The authors also insist on using the "green chemistry" argument. Sincerely, this is far for being green! For example, (l. 113, 115) formic acid is not green at all. It was used as a solvent and a building block (in excess-suppl. info l. 184). It is an inhibitor of the mitochondrial cytochrome oxidase causing histotoxic hypoxia. (Pharmacol Toxicol. 1991 Sep;69(3):157-63.) The authors criticize (l. 77) other Ti MOFs for being "non-green" as I could understand. Most use other alkoxides, and those used in the present paper are not better. The authors should not emphasize green aspects, but the remarkable stability.

Response

Following the referee's comment, we have already either changed 'green' into 'less harmful' or deleted the statements that emphasize the green aspects in the revised manuscript. And all those corresponding changes were marked in red.

Ti alkoxides are quite difficult to produce in scale, and honestly I don't think that $\text{Ti}(\text{iPrO})_4$ is cost effective (l. 92); maybe when compared to other Ti compounds, but not in a general way, for industrial purposes. Is it cheaper than TiCl_4 ?

Response

The referee was right. TiCl_4 is cheaper than $\text{Ti}(\text{iPrO})_4$ and the other Ti alkoxides in general.

In order to investigate the influence of Ti(IV) precursors on the efficiency of the MIL-177-LT MOF preparation, different Ti(IV) precursors including TiCl_4 , $\text{TiO}(\text{acac})_2$ and various Ti alkoxides were applied under the same reaction condition mentioned in the manuscript. The experimental observation supports that all the aforementioned Ti(IV) precursors work well to form MIL-177-LT compounds despite the different crystallinity of the corresponding products. Since $\text{Ti}(\text{iPrO})_4$ is less sensitive to air and easier to handle than TiCl_4 (strongly fuming and very corrosive), we selected $\text{Ti}(\text{iPrO})_4$ as the reactant for the typical synthesis of MIL-177-LT with its best compromise of cost, toxicity and easier handle.

We have deleted all the 'cost-effective' that described the chemicals and the method in the revised manuscript following the referee's comment.

The references are outdated. Many important or new papers in the field should be cited as for example : New J. Chem., 2017, 41, 14030 / C. K. Brozek and M. Dinca, J. Am. Chem. Soc., 2013, 135, 12886–12891. / J Am Chem Soc. 2016 Apr 6;138 (13):4330-3, to cite a few. This field is growing fast.

Response

The references mentioned in the comments have already been added as ref 22, 23 and 24 in the revised manuscript.

There are some minor issues too:

a: (l. 69) materials (misspelled)

Response

We have changed this item into ‘MOFs’ in the revised manuscript.

b: The authors should comment the main differences between their MOFs and MOF 901 from Yaghi et al.

Response

Besides the comparison with the other Ti-MOFs mentioned in the introduction part, we have added a conclusive part to address the difference of MIL-177 compared with the other Ti-MOFs in the conclusion part in the revised manuscript (line 289-292).

c: (l. 144) what about resistance to bases? (ammonia especially). Some figure in FIG S12, why nothing is mentioned in the main text about bases?

Response

Following this comment, a statement of the stability of MIL-177 in basic conditions (including basic aqueous condition and NH₄OH vapor) has been added into the revised manuscript (line 144-146).

d: Is there any reference for MIL-103 (la)?

Response

A corresponding reference (ref. 55) has been added for MIL-103(La) in the revised manuscript.

Reviewer #2 (Remarks to the Author):

1) Lines 210-211: The reason why the conductivity increase upon Fe doping should be explained.

Response

Following this comment, we have revised and added ‘As a typical illustration, the electrical conductive performance recorded at 1 Hz and 373 K highlights an increase from $5.2 \times 10^{-14} \text{ S.cm}^{-1}$ to $1.5 \times 10^{-10} \text{ S.cm}^{-1}$ upon Fe doping, nearly 4 orders of magnitude (See SI for the detail of synthesis, characterizations and conductive performance measurement). This change is most probably due to the fact that the iron doping is expected not only to inject free charge carries but also facilitate the charge carrier pathway.’ in the revised manuscript to explain the possible reason (line 215-219).

2) Lines 214-216: It is described that the band gap of 3.67 eV for MIL-177-HT is close to that of anatase (3.2 eV), but the difference between them 0.5 eV is not small.

Response

Following this comment, we have changed the statement as ‘This value is comparable with that for bulk anatase (3.2 eV)⁴¹, nanosized anatase (3.4-3.9 eV)⁴² and the other reported Ti-oxoclusters based MOFs such as MIL-125 (3.7 eV)⁴³ or COK-69 (3.8 eV)²¹’ and added one literature for the band gaps of nanosized anatase compounds as ref. 42 in the revised manuscript (line 223-225).

Reviewer #3 (Remarks to the Author):

1) A number of sentences throughout the manuscript are unclear, and require clarification. For example, in the introduction “Unsatisfactorily, current progress on...”; “...can offer a new line of thoughts.”; “These materilas..”; “have been hindered away from...”; “...no matter the inorganic building blocks of those Ti-MOFs are discrete Ti ions...” In general, the grammar requires attention throughout this manuscript.

Response

Following this comment, all those unclear parts in the previous manuscript have been rephrased in the revised one. The entire manuscript was edited by native speaker for improving the English.

2) References 11, 12 and 13 in the introduction do not appear to be representative of the “potential applications” mentioned. The authors should consider instead referencing one or two very recent, up-to-date, and definitive books or comprehensive reviews in the field. Readers outside the field need to be able to refer to literature that will provide an overarching view of the area.

Response

Following this comment, two recent review papers and one recent book have been added as ref. 11, 12 and 13 in the revised manuscript to replace the corresponding ones in the previous manuscript.

3) A reference is missing for “MIL-125” on page 2 of the introduction.

Response

A reference has been added for MIL-125 in the right position in the revised manuscript.

4) In general, the authors should specifically refer to the figure/table in the supplementary information that is being referred to. A couple of supplementary figures are noted throughout the article, but these do not appear to be in any particular order.

Response

Following this comment, we have rearranged the figures in the supplementary information in order to make them appear in the revised manuscript following an ascending order. Also we have referred the figure/table in the supplementary information in the revised manuscript.

5) In the Synthesis and Characterization, the authors should direct the reader to the data demonstrating “uniform size distribution” (this should also be mentioned in the main text), and “scale-up to any quantity levels” (in addition to rewording this text, mention should be made of just how scalable the synthesis is).

Response

Following this comment, Figure 2a that demonstrates ‘uniform size distribution’ of MIL-177-LT product has been referred in the revised manuscript. ‘scale-up to any quantity levels’ in the previous manuscript has been reworded into ‘scale-up to large quantity levels. For instance, 100 g of MOF product could be obtained from a single reaction in a 2 L round bottom flask for a lab scale.’ in the revised manuscript to address the comment of referee.

6) In Chemical Stability, the authors should again direct the reader to data that confirms the stability under the various conditions discussed.

Response

Following this comment, figure/table in both main text and supplementary information have been referred to direct the readers to corresponding data in the revised manuscript.

7) Defects are critical in framework chemistry with regard to functional properties such as gas storage and optical/electronic properties (eg, photoconductivity). Under Chemical Stability, the authors mention that there may be structural defects. Were these investigated and characterized? The measured properties are for the samples with the defects generated under the synthetic conditions trialled, so the inclusion of such information is important.

Response

We appreciate that the referee pointed out this important concern. The structural defects in MOFs compounds are strongly related to their property performances. However, we are lack of sufficient evidence to prove the presence of structural defects after the chemical treatment with aqua regia. So we have changed the corresponding statement and added one recent reference (ref. 29) to support this statement in the revised manuscript (line 162-163).

8) Under Figure 4, “To the best of our knowledge, increase of dimensionality of the...” is unclear and needs revision.

Response

Following this comment, ‘such as connecting the 0D clusters into 1D infinite chain or 1D chains into 2D layers through coordination bonds,’ has been added as an example to explain our statement in the revised manuscript (line 209-210).

9) In the Photoconductivity section, the measured conductivities of $<10^{-10}$ S/cm are in the insulating domain. Thus, the authors should be very cautious in their use of “electrical conductivity” and “photoconductivity”. In the abstract, the authors mention the improvement in “electrical conductive performance” with doping, however it is the photoconductive performance that is tuned.

Response

We are sorry for the confusion. On one hand, the photoconductive performance of MIL-177-HT could be adjusted by inserting polythiophene into its 1D channels in the crystal structure. And those photoconductive values are comparable with reported semi-conductive TiO₂ materials. On the other hand, the electrical conductive performance of MIL-177-HT could be changed from 5.2×10^{-14} S.cm⁻¹ to 1.5×10^{-10} S.cm⁻¹ upon Fe doping. Although both pristine and doped samples are insulators, the increase of conducting value by four orders of magnitude is clear. We have changed ‘electrical conductivity’ into ‘electrical conductive performance’ or ‘conducting’ in the revised manuscript.

10) In the Photoconductivity section, “Secondly, first...” should be fixed.

Response

Thank you for your pointing it out. We fixed as ‘Secondly,...’ in the revised manuscript.

11) References 42, 43 and 44 in the Photoconductivity section are for electrical conduction in MOFs, not photoconductivity. The authors need to instead reference specific works in the MOF field that report photoconductivity.

Response

We are sorry for this unclear statement and confusion. We have changed the statement into ‘Recently, conductive MOFs have attracted considerable attention, however, this research field mainly focused on the electrically conductive ones^{46,47,48}’ in the revised manuscript (line 244-245) in order to make it clear for readers.

12) Supporting Information Figure S2 – what is the origin of the indentation at ~450 deg C in the plot for MIL-177-HT?

Response

The indentation occurred at around 450 °C in the TGA curve of MIL-177-HT is due to the instrument turbulence. Thus following the comment, we have already recollected the TGA data on the MIL-177-HT sample and included the corresponding TGA figures in the revised SI.

REVIEWERS' COMMENTS:

Reviewer #3 (Remarks to the Author):

The responses provided by Serre and coworkers to the referees comments are suitable and I recommend publication subject to some minor corrections:

- 1) Figure 2 - clearly add the scale-bar for the SEM image in figure a.
- 2) Line 245 (addition of sentence and references for conductive MOFs) - the authors should consider referencing a more recent review article in this area for which there are a number. E.g. the special issue of MRS Bulletin - Volume 41, Issue 11 (Metal–Organic Frameworks for Electronics and Photonics) p. 858 etc.
- 3) In general, there are a number of grammatical errors remaining that should be attended to in the final editorial process.

Response to Reviewers' comments:

Reviewers' comments:

Reviewer #3 (Remarks to the Author):

The responses provided by Serre and coworkers to the referees comments are suitable and I recommend publication subject to some minor corrections:

1) Figure 2 - clearly add the scale-bar for the SEM image in figure a.

Response:

We have added a scale-bar for the SEM image in Figure 2a in the revised main text according to the referee's comment.

2) Line 245 (addition of sentence and references for conductive MOFs) - the authors should consider referencing a more recent review article in this area for which there are a number. E.g. the special issue of MRS Bulletin - Volume 41, Issue 11 (Metal–Organic Frameworks for Electronics and Photonics) p. 858 etc.

Response:

We have added two review papers mentioned by the comment as ref.50 and 51 in the revised main text.

3) In general, there are a number of grammatical errors remaining that should be attended to in the final editorial process.

Response:

Thank you for bringing this to our attention. Please find the revised manuscript in the uploaded files.